# Effects of the Physician–Primary-Healthcare Nurse Telemedicine Model (P-NTM) on Medication Adherence and Health-Related Quality of Life (HRQoL) of Patients with Chronic Disease at Remote Rural Areas

**DOI:** 10.3390/ijerph18052502

**Published:** 2021-03-03

**Authors:** Mi Young Kwak, Eun Jeong Hwang, Tae Ho Lee

**Affiliations:** 1Center for Public Healthcare, National Medical Center, 245 Euljiro, Jung-gu, Seoul 04564, Korea; kmy805@gmail.com (M.Y.K.); leeth95@naver.com (T.H.L.); 2Department of Nursing, Sehan University, 1113, Samho-eup, Yeongam-gun, Jeollanam-do 58447, Korea

**Keywords:** chronic-disease management, medication adherence, health-related quality of life, telemedicine

## Abstract

Chronic diseases are a major cause of death and have a negative impact on community health. This study explored the effects of a chronic-disease management program utilizing the physician–primary-healthcare nurse telemedicine model (P–NTM) on medication adherence and health-related quality of life (HRQoL) in 113 patients with chronic diseases in remote rural areas. We used a quasi-experimental, nonequivalent-control-group pretest–post-test design. This study used secondary data from the 2018 Pilot Telemedicine Project for Underserved Remote Rural Areas. In this study, 113 subjects participated, in which the patient’s first visit was assigned as a control group for the previous face-to-face hospital care; after three months of receiving the P–NTM program, the same subjects were assigned to be the experiment group for P–NTM. Data were analyzed by using descriptive statistics, a paired *t*-test, and logistic regression. With regard to the results, subjects showed a 1.76 times higher probability of improving medication adherence after participating in P–NTM compared to hospital care (odds ratio (OR) = 1.76, 95% confidence interval (CI) = 1.34–2.31). Our findings showed that patients with chronic diseases, especially those who reside in remote rural areas, should be provided with effective health services, utilizing various strategies to enhance a healthy life.

## 1. Introduction

Chronic diseases are the leading global cause of death, accounting for about 70% of all deaths [1]. Most chronic diseases may deteriorate the living capacity and functional levels of patients, thereby reducing the overall health level and health-related quality of life (HRQoL) [2]. If patients with chronic diseases do not follow medication guidelines, they cause negative clinical results, and HRQoL is lowered, causing additional costs [2,3,4]. Several previous studies [3,4,5,6,7,8,9,10,11] reported that a major problem with chronic diseases is that it is difficult to maintain proper disease self-management for a long time, such as medication adherence. As many as 30–50% of chronic-disease patients reported that they do not take drugs as prescribed [9,10]. The rate of medication adherence among patients with chronic diseases has been diversely reported to be 30–70% for hypertension patients [7] and 36–93% for type 2 diabetes patients [8]. In particular, it may be more difficult for older adults due to cognitive decline. It was reported that 38.8% of the elderly forget their medication, and 14.3% complained about the difficulty of medication [11]. In general, chronic diseases progress slowly over a long period of time, so the prevalence of chronic diseases tends to exponentially rise with age, including in Korea. It was reported that 87.1% of those aged 65 years or older suffer from one or more chronic diseases, and that 18% of those suffer both hypertension and diabetes in Korea [12]. Moreover, patients with chronic diseases reside in remote rural areas that are greatly limited in hospital accessibility, so there is a high possibility that greater problems occur in the treatment of diseases and HRQoL. The lack of specialty-care services, physician shortage, and geographic separation in remote rural areas are particularly problematic for patients with chronic diseases [13]. In particular, rural areas may have more problems in the management of patients with chronic diseases due to the higher proportion of elderly populations. The HRQoL of patients with chronic diseases improves if they are provided with early treatment for chronic diseases, steady intervention for behavioral changes, and psychological-disorder intervention [2]. The World Health Organization (WHO) emphasized [10] that innovations to improve treatment adherence may contribute more to public health than developing specific medical treatments. In the last few decades, telemedicine has been conducted to effectively treat and manage patients with chronic diseases who reside in remote rural areas.

Several previous studies [13,14,15,16,17] reported that telemedicine has contributed to increased access to and improved quality of medical services, particularly in remote rural areas. Telemedicine has obvious advantages in remote areas where it improves access to health services, obviating the need for patients, their caregivers, and healthcare workers to travel [13]. Mehrotra et al. [15] reported that telemedicine subjects increased by more than 25% each year for 10 years, and that there is improved medical accessibility to sick and poor people in rural areas. Remote monitoring through telemedicine is effective for chronic diseases such as diabetes [18], depression [19], and dementia [14]. Wootton [16], reviewing studies related to telemedicine published in the past 20 years, identified that telemedicine had a great effect on chronic-disease management. In addition, telemedicine is suitable for older patients and aims for the early detection and rapid treatment to reduce health deterioration and maintain function and independence [19]. Among older populations in remote rural areas who are at increased risk for chronic diseases, telemedicine has the potential to improve access to healthcare services, thus perhaps reducing adverse health outcomes [13]. Although telemedicine has the limitation that it cannot facilitate smooth interactions between doctors and patients, it can improve the accessibility and quality of patients in areas with poor medical access or rural areas without direct contact with physicians [13,15,16]. In Korea, the medical law on telemedicine was passed in 2002 to provide only to underserved people who reside remote rural areas. On a legal basis, the physician–primary-healthcare nurse telemedicine model (P–NTM) was developed [20]. The P–NTM is a method that manages patients with chronic diseases residing in underserved remote rural areas by interacting with a distant physician and a primary-healthcare nurse at the local site using information and communication technology.

Chronic diseases exist in a patient’s life for a long time, and they greatly affect their way of life. Therefore, it is necessary to apply a variety of appropriate management strategies over a long time to follow prescribed treatments, continuously control and manage symptoms, and help maintain an optimal life, even in sickness. Patients with chronic disease living in remote areas should not be an exception from effective chronic-disease management. Korean government applied various pilot projects including telemedicine for patients with chronic diseases living in remote areas, but few studies systematically verified the effectiveness of the projects. It is necessary to empirically verify the effects of the chronic-disease management remote-treatment pilot project on the medication adherence and HRQoL of patients with chronic disease in remote rural areas. Therefore, this study explored the effects of a chronic-disease management program utilizing the P–NTM on medication adherence and HRQoL in patients with chronic diseases at remote rural areas.

## 2. Materials and Methods

### 2.1. Design

The study was conducted by using a quasi-experimental, nonequivalent-control-group pretest–post-test design.

### 2.2. Data Collection and Procedures

In this study, secondary data, which are the data of the 2018 Telemedicine Pilot Project at Remote Rural Areas, were used [20]. The subjects of this study were patients with hypertension, hyperlipidemia, and diabetes visited in primary-healthcare centers or clinics located in 11 remote rural areas selected as medically underserved locations. The minimal sample size required for two-tailed *t*-test analysis was determined by using G*power software (version 3.0) (Faul, F.; Erdfelder, E.; Lang, A.-G.; Buchner, Olshausenstr. 40, D-24098 Kiel, Germany) and setting the effect size at 0.8, significance level (α) at 0.05, and power (1 − *β*) at 80%. The minimal sample size required was calculated to be 102 individuals [21]. Lastly, 113 subjects participated in this study, so the number of subjects in this study met the minimum. They consisted of subjects who had already been treated for chronic diseases in hospitals before participating in P–NTM. Data on the first day of the visit were collected for medication adherence and HRQoL for the previous face-to-face hospital care. In addition, medication adherence and HRQoL related to telemedicine were repeatedly collected after three months of participation in the P–NTM. Medication adherence and HRQoL were collected on the basis of treatment experiences received from hospitals prior to telemedicine and set as controls, and medication adherence and HRQoL were collected on the basis of experience three months after participation in P–NTM. Quality was set as the experiment group. Before the study was conducted, subjects had had the study explained to them and given their consent to participate in the study. The study process is shown in Figure 1.

#### 2.2.1. Chronic-Disease Management Program Utilizing P–NTM

The Korean government has been conducting pilot telemedicine projects to manage chronic diseases only in remote rural areas that are medically underserved locations in accordance with the medical laws created in 2002. In the law, telemedicine is defined as “providing medical knowledge and technology to a distant doctor (physicians, dentists, oriental-medicine doctors) by using medical information and communication technologies such as computers and video communications” [22]. The served areas of telemedicine are designated to be where there is poor access to health and medical resources, poor-quality medical services, and low levels of health and health outcomes [20], so the physician–primary-healthcare nurse telemedicine model (P–NTM) was developed [20]. This model has been developed concerning informed consent, protection data, confidentiality, physician’s malpractice, and liability and telemedicine regulations, which were suggested as ethical aspects of telemedicine [23]. In this study, the P–NTM is a pilot telemedicine service for chronic patients who reside underserved rural areas. A primary-healthcare nurse sends referrals and clinical information about a patient with a chronic disease to a remote physician using information and communication technology (computers, video communications, etc.). Then, the physician answers with prescription and medical-treatment recommendations for the patients to the nurse using information and communication technology. A distant physician at a hospital or a public healthcare center provides medical expertise and opinions to the local primary-healthcare nurse at the remote health clinic using information and communication technology. On the basis of the opinions of the distant physician, the local nurse provides counseling, physical examination, treatment, and medication to the local patients with chronic diseases. A distant physician may provide diagnosis and prescription to local patients with chronic diseases when necessary.

#### 2.2.2. Medication Adherence

We used the Korean version of the modified Morisky scale (MMS) [24], developed by the Case Management Society of America (CMSA) [25]. The MMS was developed by supplementing the potential deficiencies of the original Morisky Scale [26], which was designed to assess the level of knowledge and motivation related to the medication adherence of patients with hypertension [25]. The MMS is also used to measure adherence to various chronic diseases such as patients with diabetes and dyslipidemia, as well as hypertension [24,27]. The scale consisted of standardized six items to measure the knowledge and motivation of patients related to taking medicine. To measure the patient’s knowledge of taking medicine, Questions 3–5 are applicable. The motivation for taking medication is to measure the patient’s intention to adherence medication, and Questions 1, 2, and 6 were applicable. The contents of each item in this tool are: (1) Do you ever forget to take your medicine? (2) Are you careless at times about taking your medicine? (3) When you feel better do you sometimes stop taking your medicine? (4) Sometimes if you feel worse when you take your medicine, do you stop taking it? (5) Do you know the long-term benefits of taking your medicine as told to you by your doctor or pharmacist? (6) Sometimes, do you forget to refill your prescription medicine on time? Question 5 was analyzed by inverse calculation. Each item was measured on a two-point scale (1 = yes, 2 = no), with a higher score indicating higher medication adherence by the subject. Cronbach’s ⍺ of this instrument in the study was 0.65, indicating acceptable internal consistency.

#### 2.2.3. Health-Related Quality of Life (HRQoL)

We used the Korean version of the health-related quality of life (HRQoL) [28], which consisted of five items: mobility, self-care, usual activities, pain and discomfort, and anxiety and depression. Each item was measured on a three-point scale (1 = extreme problem, 2 = moderate problem, 3 = no problem), with higher scores indicating better levels of health. Cronbach’s ⍺ of this instrument in the study was 0.81, indicating acceptable internal consistency.

### 2.3. Ethical Considerations

Before conducting this study, approval was obtained from the institutional review board of the clinical examination committee of Sehan University (SH-IRB 2020-05-26-01). Participants were provided with an explanation of the proposed study, and informed that they were free to withdraw from the study at any time, without prejudice. Written informed consent was obtained from each individual. Identification data of study subjects were kept separately at the health centers and not made available to those conducting the study. P–NTM for subjects in this study was provided free of charge because the program was a pilot project by the government.

### 2.4. Statistical Analyses

Data were analyzed by using IBM SPSS version 21 software (IBM Corporation, Armonk, NY 10504, USA). Comparisons between the experiment and control groups’ medication adherence and HRQoL levels before and after the P–NTM were performed by using a paired *t*-test. Logistic regression was performed to determine independent factors associated with P–NTM.

## 3. Results

### 3.1. Sample Characteristics

The characteristics of this study’s subjects are shown in Table 1. The final participants of this study were a total of 113 patients with chronic diseases. There were 40 (35.4%) males and 73 (64.6%) females. Average age was 69.74 years (± 9.36), and the age range was 47 to 86 years. Regarding employment status, 55 (50.9%) subjects were employed, and 53 (49.1%) subjects were unemployed. Regarding health security, 106 subjects (94.6%) had national health insurance. The number of households with the highest proportion was two. Regarding disease, 75 subjects (66.4%) had hypertension, 10 subjects (8.8%) had diabetes, six subjects (5.3%) had hyperlipidemia, and 22 subjects (19.5%) had more than two chronic diseases.

In this study, the results of the Spearman’s rho analysis among the subjects’ characteristics were as follows. As results, there was a positive correlation between age and employment status (*r* = 0.54, *p* < 0.001), a positive correlation between employment status and having hypertension (*r* = 0.20, *p* = 0.037), a negative correlation between age and number of households (*r* = −0.55, *p* < 0.001), and a negative correlation between employment status and number of households (*r* = −0.43, *p* < 0.001).

### 3.2. Comparison of Variables between Previous Hospital Care and P–NTM

The comparison results of medication adherence and HRQoL between previous hospital care and P–NTM by subject characteristics are shown in Table 2. Both male (*p* = 0.003) and female (*p* < 0.001) subjects showed significant differences in medication adherence between previous hospital care and P–NTM. Only female subjects showed a significant difference in HRQoL between previous hospital care and P–NTM (*t* = −2.84, *p* = 0.006). Subjects under the age of 74 showed significant differences in medication adherence between previous hospital care and P–NTM (*p* < 0.01). Subjects aged 65 to 79 years showed significant differences in HRQoL between previous hospital care and P–NTM (*p* < 0.05). Subjects aged 80 years and older showed no significant differences in medication adherence and HRQoL between previous hospital care and P–NTM. 

### 3.3. Logistic-Regression Analyses

The logistic-regression results of the medication adherence and HRQoL of patients with chronic diseases who participated in P–NTM are shown in Table 3. Logistic-regression results in Table 4 are of the medication adherence and HRQoL of patients with a chronic disease who participated in P–NTM according to each item of the two scales. Two models, (−2 Log L = 288.21, chi-squared = 22.31, *p* < 0.001) and (−2 Log L = 276.97, chi-squared = 35.55, *p* < 0.001), met the convergence criteria for logistic regression. As a result of logistic-regression analysis, significant factors were as follows. Compared to previous face-to-face hospital care, the medication adherence of subjects was improved by 1.76 times after participation in the P–NTM (odds ratio (OR) = 1.76, 95% confidence interval (CI) = 1.34–2.31). With regard to the results of logistic-regression analysis of the detailed items of medication adherence and HRQoL, significant factors were as follows. Compared to previous face-to-face hospital care, medication-adherence motivation of subjects was improved by 2.08 times after participation in the P–NTM (OR = 2.08, 95% CI = 1.47–2.97). Compared to previous face-to-face hospital care, anxiety and depression of subjects was improved 2.25 times after participation in the P–NTM (OR = 2.25, 95% CI = 1.14–4.45). 

## 4. Discussion

This study investigated the effects of a chronic-disease management program utilizing P–NTM on the medication adherence and HRQoL of patients with chronic diseases in remote rural areas. In the results of the present study, female subjects showed a significant improvement in both medication adherence and HRQoL after participating in telemedicine, indicating that telemedicine was more effective than for male subjects with improved medication adherence. In the results of the present study, the effectiveness of P–NTM participation varied according to the age group of the subjects. In younger subjects under the age of 64, the P–NTM may have contributed to improved medication adherence, but it did not affect HRQoL. On the other hand, in subjects aged 75 to 79, P–NTM contributed to improved HRQoL but did not affect medication adherence. In subjects aged 65 to 74, P–NTM contributed to both medication adherence and HRQoL. P–NTM, however, did not affect either medication adherence or HRQoL in subjects aged 80 and older. A study by Kim et al. [29] found that subjects who received an information-technology-based glucose-monitoring service had their drug compliance increased with age. Unfortunately, the study [29] was divided into ages below 50, 50s to 60s, and over 60s, and the results did not appear significant, making it difficult to compare with the results of this study. Dal Bello-Haas et al. [14] hypothesized that older patients, whose ability to learn new technology is lower, might have more trouble using the smartphone application than younger patients [14]. Nevertheless, telemedicine was reported to be useful to elderly patients with chronic diseases, because it allows doctors to offer medical care without moving [14,19,30,31,32].

In the results of the present study, the medication adherence, especially motivation, of subjects after participating in the P–NTM was significantly improved compared to previous face-to-face hospital care. This was similar to findings of Benson et al. [33] who stated that diabetic patients receiving a telemedicine program had significantly greater medication adherence compared to the control group. According to Kim [24], elderly patients with chronic diseases have a high knowledge of medication adherence but low motivation for medication adherence. Kim [24] identified that elderly patients who had received sufficient explanation from doctors or pharmacists reported higher motivation for medication than those who had received insufficient explanations. Telemedicine has the advantage of allowing patients and doctors to interact in a comfortable, unhurried environment [13]. This explains why the subjects’ motivation for medication adherence improved significantly after participation in telemedicine in the present study.

In the results of the present study, the anxiety and depression of subjects after participating in the P–NTM was significantly improved compared to during the previous face-to-face hospital care. This was similar with the findings of Gellis et al. [19] that integrated telehealthcare with a physician and nurse for older adults with chronic illness and comorbid depression can reduce symptoms in home health settings. According to Park et al. [34], HRQoL decreases in patients with chronic diseases, accompanied by depression. According to Horvat et al. [3], the more medications to be administered and the less education, the lower the HRQoL of patients with chronic diseases. In this regard, it is a very meaningful result in the present study that patients with chronic diseases had significant improvement in depression after participating in P–NTM. In the study results of von Storch et al. [18], the self-care abilities of diabetic patients showed significant improvement after participating in telemedicine; however, in the results of the present study, there was no significant difference in self-care activity of subjects after participating in the P–NTM, so there was no consistency. Hiratsuka et al. [13] stated that the biggest advantage of telemedicine is that the movement of patients or doctors is minimized, and medical costs are accordingly reduced. Accessibility to and waiting time for medical treatment are increased and reduced, respectively [13]. Maresca et al. [31] reported that telemedicine can be a suitable tool for more efficiently caring for elderly people by promoting the remission of depressive symptoms and improving social functioning, cognitive levels, and nutritional habits to prevent vascular diseases and the exacerbation of pre-existing chronic illness. Telemedicine showed many possibilities in managing chronic diseases in remote medical vulnerabilities [15]. The high turnover of providers in rural areas and the lack of specialty-care services throughout remote rural healthcare-delivery areas have led to a growing interest in telemedicine-supplemented health systems [13]. Telemedicine programs are promoted as a solution to improving access to healthcare for rural, remote, and medically underserved communities [13]. Chronic diseases cannot be easily treated and need to be steadily managed for a long time. Due to the nature of these chronic diseases, effectively improving access to medical personnel can be key to disease management. Chronic-disease patients in rural and medically vulnerable destinations can go to doctors and receive face-to-face medical treatments, which can be difficult beyond inconvenience. Therefore, it is necessary to apply a variety of appropriate management strategies, such as telemedicine, for a long time to follow prescribed treatments, continuously control and manage symptoms, and support the maintenance of an optimal life, even in a diseased state. According to Farrell [30], seniors that received telehealth showed a decrease in loneliness, isolation, and depression, as well as a reduction in hospital admissions and readmissions. He [30] stated that telehealth increases patients’ quality of life and reduces depression by allowing patients to live at home while being able to effectively and inexpensively receive treatment. Through previous studies and the results of the present study, telemedicine may be effective in managing the chronic diseases of patients. On the other hand, although telemedicine has significantly improved the accessibility of medical care to the poor and sick in rural areas, it is uncertain whether it has helped improve their health [15]. Some previous studies have suggested some implications related to telemedicine for patients with chronic diseases residing in remote rural areas. In the study by Hiratsuka et al. [13], subjects suggested that the problem of telemedicine is that video data are not clear, there are technical problems such as not being able to connect to the internet, and there are obstacles due to the lack of face-to-face contact. In addition, the study [13] suggested to establish a relationship between patient and doctor in telemedicine and emphasized that the first doctor’s meeting should be face-to-face, and that the same doctor should continue to do the treatment in telemedicine. In addition, many problems, such as legal and ethical issues, insufficient software, prescription issuance, and storage methods have emerged for telemedicine [23,35], and it may be necessary for stakeholders to co-ordinate their opinions. In the event of a medical accident, it is necessary to predict various problems, such as liability, and prepare alternatives so that telemedicine can successfully settle [36]. Since telemedicine is more likely to be gradually expanded in the current era when information and communication technology is developed, and a new medical-delivery system is required, when telemedicine is introduced in the future, the subject of telemedicine application, eligibility criteria, and insurance benefits should be applied. Face-to-face treatment may become more difficult as various large scale infectious diseases, such as COVID-19, etc. [37]. In addition, the demand for telemedicine is expected to increase further for the provision of medical care in rural areas with low access to hospitals.

### 4.1. Limitations

The present study had several limitations. All collected secondary data were cross-sectional, making it difficult to make causal inferences. The data of the present study were collected by eleven healthcare nurses at 11 healthcare clinics, respectively. Although data were collected using the same structured questionnaire, there could have been exogenous variable intervention, because subject characteristics and environmental factors could work in various ways. Attempts to generalize the results of this study, which were obtained using secondary data originally collected for another purpose, must be undertaken with caution. The acquired data have inherent limitations. For example, we did not have information on whether subjects had a history of major depression. The data used in this study were collected with self-reporting. Thus, the possibility of response bias cannot be eliminated.

### 4.2. Implications for Further Research

There is a need for continuing studies that consider subject characteristics, for example, the severity of depression or other diseases. Additionally, we suggest studies applying P–NTM for various diseases such as dementia and arthritis. We also propose studies that analyze the impact of P–NTM on patients as well as their families or caregivers.

## 5. Conclusions

This study investigated the effects of a chronic-disease management program utilizing the P–NTM on the medication adherence and HRQoL of patients with chronic diseases at remote rural areas. Chronic diseases are not easily curable, and it is necessary to apply a variety of appropriate management strategies over a long time to follow prescribed treatments, continuously control and manage symptoms, and to help maintain an optimal life, even in sickness. It may be expected that telemedicine will be expanded not only for patients who reside in underserved areas, but also for patients with chronic diseases in cities, who can stay at home and receive cost-effective treatment. Moreover, face-to-face care in hospitals will inevitably shrink due to infections in hospitals and the continuous management of infectious diseases.

## Figures and Tables

**Figure 1 ijerph-18-02502-f001:**
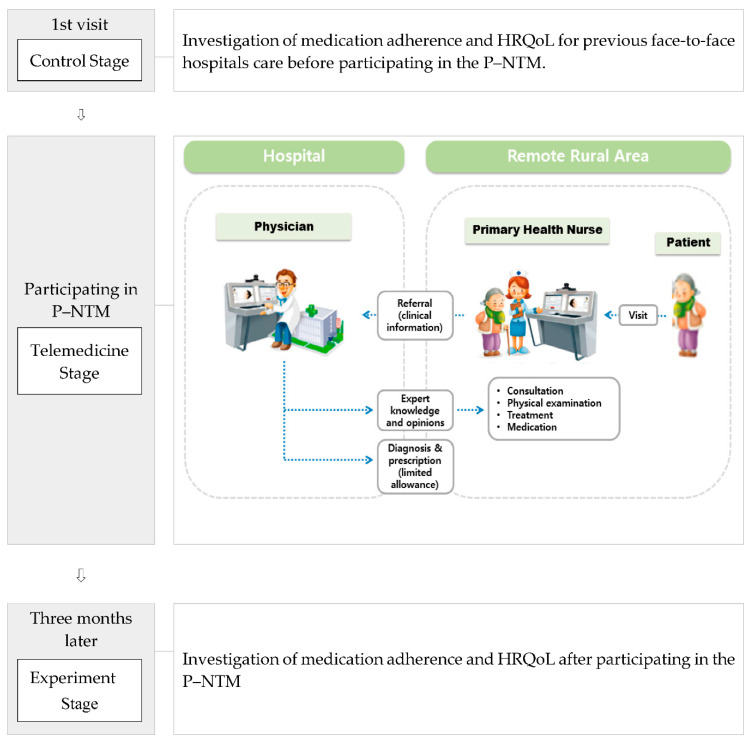
Process of this study, including physician–primary-healthcare nurse telemedicine model physician–primary-healthcare nurse telemedicine (P–NTM). HRQoL, health-related quality of life.

**Table 1 ijerph-18-02502-t001:** Subject characteristics (N = 113) ^1^.

Characteristics	Categories	n	%
Gender	Male	40	35.4
	Female	73	64.6
Age (years)	≤64	34	31.5
	65–69	16	14.8
	70–74	16	14.8
	75–79	22	20.4
	≥80	20	18.5
	M ± SD	69.74 ± 9.36	
	Range	47–86	
Employment status	Employment	55	50.9
	Unemployment	53	49.1
Health security	National health insurance	106	94.6
	Medical benefits	6	5.4
No. of households	Living alone	43	38.1
	Two households	59	52.2
	Over three households	11	9.7
Disease	Hypertension	75	66.4
	Hyperlipidemia	6	5.3
	Diabetes mellitus	10	8.8
	More than two diseases	22	19.5

^1^ Missing data: age (*n* = 5); employment status (*n* = 5), health security (*n* = 1).

**Table 2 ijerph-18-02502-t002:** Comparison of medication adherence and HRQoL between previous hospital care and P–NTM by subject characteristics ^1^.

Characteristics	Items	Medication Adherence			HRQoL ^2^		
		Previous Hospital Care	P−NTM ^3^	*t* (*p*)	Previous Hospital Care	P−NTM ^3^	*t* (*p*)
Gender	Male	10.70 ± 1.11	11.23 ± 0.89	**−3.20 (0.003)**	14.13 ± 1.18	14.35 ± 1.23	−1.94 (0.060)
	Female	10.25 ± 1.79	11.20 ± 0.95	**−4.21 (<0.001)**	13.24 ± 1.62	13.49 ± 1.74	**−2.84 (0.006)**
Age, (years)	≤64	10.27 ± 2.25	11.35 ± 0.73	**−2.76 (0.009)**	14.44 ± 0.96	14.47 ± 1.08	−0.30 (0.768)
	65–69	9.88 ± 1.31	11.00 ± 0.97	**−3.74 (0.002)**	13.75 ± 1.13	14.38 ± 1.09	**−4.04 (0.001)**
	70–74	10.31 ± 1.20	11.19 ± 0.83	**−3.96 (0.001)**	13.19 ± 1.97	13.75 ± 2.08	**−2.76 (0.014)**
	75–79	10.90 ± 1.04	11.33 ± 0.80	−1.63 (0.119)	13.10 ± 1.41	13.43 ± 1.60	**−2.09 (0.049)**
	≥80	10.65 ± 0.99	10.90 ± 1.33	−0.84 (0.412)	12.65 ± 1.66	12.70 ± 1.69	−0.25 (0.804)
Employment status	Employment	10.25 ± 1.93	11.09 ± 0.91	**−3.12 (0.003)**	14.33 ± 0.92	14.51 ± 0.90	**−2.63 (0.011)**
	Unemployment	10.57 ± 1.12	11.37 ± 0.85	**−4.92 (<0.001)**	12.71 ± 1.65	13.12 ± 1.90	**−3.44 (0.001)**
Health security	National health insurance	10.41 ± 1.61	13.57 ± 1.54	**−4.94 (<0.001)**	13.57 ± 1.54	13.84 ± 1.59	**−3.29 (0.001)**
	Medical benefits	10.83 ± 1.17	12.67 ± 1.97	−1.20 (0.286)	12.67 ± 1.97	12.83 ± 2.14	−0.54 (0.611)
Number of households	Living alone	10.79 ± 0.92	11.17 ± 1.08	−1.95 (0.058)	13.07 ± 1.55	13.33 ± 1.65	**−2.05 (0.047)**
	Two households	10.07 ± 1.96	11.25 ± 0.84	**−4.80 (<0.001)**	13.81 ± 1.50	14.12 ± 1.55	**−3.34 (0.001)**
	Over three households	10.90 ± 0.74	11.10 ± 0.74	−1.50 (0.168)	14.00 ± 1.26	13.82 ± 1.60	1.49 (0.167)
Disease	Hypertension	10.42 ±1.22	11.21 ± 0.87	**−5.92 (<0.001)**	13.49 ± 1.55	13.73 ± 1.70	**−2.77 (0.007)**
	Hyperlipidemia	10.83 ± 0.98	11.17 ± 0.98	−1.00 (0.363)	13.33 ± 2.73	13.50 ± 2.35	−1.00 (0.363)
	Diabetes	10.00 ± 3.59	11.70 ± 0.67	−1.41 (0.191)	14.10 ± 0.88	14.60 ± 0.70	**−3.00 (0.015)**
	Over two diseases	10.45 ± 1.47	11.00 ± 1.16	−1.67 (0.110)	13.59 ± 1.33	13.72 ± 1.42	−0.77 (0.451)

^1^ Missing data: age (*n* = 5); employment status (*n* = 5), health security (*n* = 1). ^2^ HRQoL = health-related quality of life. ^3^ P–NTM = physician–primary-healthcare nurse telemedicine model to treat patients with chronic diseases. Bold numbers indicate significant values.

**Table 3 ijerph-18-02502-t003:** Logistic-regression model for medication adherence and HRQoL of patients with chronic disease who participated in P–NTM.

Variables	B	S.E.	*p*	Exp (B)	95% CI
Medication adherence	**0.57**	**0.14**	**<0.001**	**1.76**	**1.34–2.31**
HRQoL	0.07	0.09	0.459	1.07	0.90–1.27
Constant	−7.07	1.85	0.000	0.001	

OR = odds ratio. CI = confidence interval. Bold numbers indicated significant values.

**Table 4 ijerph-18-02502-t004:** Logistic-regression model for medication adherence and HRQoL of patients with chronic disease who participated in P–NTM according to detailed items of each scale ^1^.

	B	S.E.	*p*	Exp (B)	95% CI
Knowledge of medication adherence	0.22	0.26	0.408	1.24	0.74–2.07
**Motivation of medication adherence**	**0.73**	**0.18**	**0.000**	**2.08**	**1.47–2.97**
Mobility	0.34	0.52	0.513	1.41	0.54–3.93
Self–care	−0.53	0.59	0.371	0.59	0.19–1.88
Usual activity	−0.84	0.53	0.113	0.43	0.15–1.22
Pain and discomfort	0.40	0.33	0.224	1.49	0.78–2.84
**Anxiety and depression**	**0.81**	**0.35**	**0.019**	**2.25**	**1.14–4.45**
Constant	−5.40	2.19	0.014	0.01	

^1^ Bold numbers indicate significant values.

## Data Availability

Restrictions apply to the availability of these data. Data was obtained from the National Medical Center (NMC) and are available from the authors with the permission of the NMC.

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
