# Peer review of "Effects of the Physician–Primary-Healthcare Nurse Telemedicine Model (P-NTM) on Medication Adherence and Health-Related Quality of Life (HRQoL) of Patients with Chronic Disease at Remote Rural Areas"

_ijerph, 2021, doi:10.3390/ijerph18052502_

Round 1

Reviewer 1 Report

POINT 1

The authors managed  to partially evaluate all the implications related to the topic developed in this work; implementation of  Telemedicina  to improve the quality of life of patients with chronic diseases in rural areas. This is, in general, an important contribution because this topic is one of the serious public health problems of the world. The originality of this paper may inform the need for new strategies in order to develop preventative measures, but I have some major suggestions on this article. In general, the paper was well summarized and evaluated, but in the introduction section, the aim of paper in terms of why the paper is written and what knowledge gap is being addressed or identified was not clearly stated. Therefore, the authors should address the aim of the paper highlighting it in the same introduction. This would indicate or add to the significance of this work thereby adding to the literature in the area.

POINT 2

In the introduction section lines 72 and 73, the authors mentioned of the ethical implications, referring to the doctor-patient relationship, a very important topic in  telemedicine that must be expanded with greater attention, analyzing  authorities and professional responsibilities in telemedicne and telehealth procedures. This is a very important issue that has been addressed superficially by the authors. A deepening of the ethical study and professional responsibility issue must be addressed by the authors. I suggest to the authorssome recent works that should be analyzed and integrated; a recent review of the literature entitled "Telemedicine Practice: Review of the Current Ethical and Legal Challenges” recently published on Telemed J E Health journal. Doi: 10.1089/tmj.2019.0158 and the article " Consequences of COVID-19 Outbreak in Italy: Medical Responsibilities and Governmental Measures" Front Public Health. 2020 Dec 8;8:588852. doi: 10.3389/fpubh.2020.588852.

POINT 3

In the final paragraph of the introduction section from line 81 to line 87, the authors refer to innovative management strategies for the management of chronic diseases. I believe the authors should extend this session by referring to recent works in the literature such as "A pilot study on the management of patients with insulin pumps during the COVID-19 pandemic. Diabetes Res Clin Pract. 2020 Nov; 169: 108481. Doi: 10.1016 / j.diabres.2020.108481.

POINT 4

The authors, in this paper, face a very important problem that concerns the monitoring and the correct use of the drugs. This is a fundamental concept for the good result of a drug therapy.  The authors should emphasize and investigate the issue of monitoring and drug therapy in support of Telemedicine implementations. A study addressing this problem entitled “Evaluation of medical prescriptions and off-label use ..... to improve healthcare quality ", published in the Eur Rev Med Pharmacol Sci. Doi: 10.26355 / eurrev_201807_15439, should be considered by the authors to discuss the problem.

POINT 5

English grammar, punctuation, and spelling need to be corrected, and native English revision is required.

POINT 6

Strength: in general, the study is easy to follow and well summarized, raising the main issues related to the topic. Limitation: although the main issues related to the topic raised by the authors, the report is not well interpreted in terms of the issue raised and the purpose/aim of the review is not clearly illustrated, related to other literature on the topic not well mentioned, in particular, recent studies have not taken into consideration.

Author Response

Red letters in the manuscript indicate revised sentence.

POINT 1

The authors managed to partially evaluate all the implications related to the topic developed in this work; implementation of Telemedicina to improve the quality of life of patients with chronic diseases in rural areas. This is, in general, an important contribution because this topic is one of the serious public health problems of the world. The originality of this paper may inform the need for new strategies in order to develop preventative measures, but I have some major suggestions on this article. In general, the paper was well summarized and evaluated, but in the introduction section, the aim of paper in terms of why the paper is written and what knowledge gap is being addressed or identified was not clearly stated. Therefore, the authors should address the aim of the paper highlighting it in the same introduction. This would indicate or add to the significance of this work thereby adding to the literature in the area.

  • We revised according to reviewer’s comment. Please refer to the red color sentence on page 2, lines 82-85.

POINT 2

In the introduction section lines 72 and 73, the authors mentioned of the ethical implications, referring to the doctor-patient relationship, a very important topic in telemedicine that must be expanded with greater attention, analyzing authorities and professional responsibilities in telemedicne and telehealth procedures. This is a very important issue that has been addressed superficially by the authors. A deepening of the ethical study and professional responsibility issue must be addressed by the authors. I suggest to the authorssome recent works that should be analyzed and integrated; a recent review of the literature entitled "Telemedicine Practice: Review of the Current Ethical and Legal Challenges” recently published on Telemed J E Health journal. Doi: 10.1089/tmj.2019.0158 and the article " Consequences of COVID-19 Outbreak in Italy: Medical Responsibilities and Governmental Measures" Front Public Health. 2020 Dec 8;8:588852. doi:10.3389/fpubh.2020.588852.

  • We revised according to reviewer’s comment. Please refer to the red color sentence on page 3, lines 122-124.

POINT 3

In the final paragraph of the introduction section from line 81 to line 87, the authors refer to innovative management strategies for the management of chronic diseases. I believe the authors should extend this session by referring to recent works in the literature such as "A pilot study on the management of patients with insulin pumps during the COVID-19 pandemic. Diabetes Res Clin Pract. 2020 Nov; 169: 108481. Doi: 10.1016 / j.diabres.2020.108481.

  • Thank you for your suggestions. However, the literature you have suggested is considered to be less relevant to this study. We will apply it to the next opportunity.

POINT 4

The authors, in this paper, face a very important problem that concerns the monitoring and the correct use of the drugs. This is a fundamental concept for the good result of a drug therapy.  The authors should emphasize and investigate the issue of monitoring and drug therapy in support of Telemedicine implementations. A study addressing this problem entitled “Evaluation of medical prescriptions and off-label use ..... to improve healthcare quality ", published in the Eur Rev Med Pharmacol Sci. Doi: 10.26355 / eurrev_201807_15439, should be considered by the authors to discuss the problem.

  • Thank you for your suggestions. However, the literatures you have suggested are considered to be less relevant to this study. We will apply them to the next opportunity.

POINT 5

English grammar, punctuation, and spelling need to be corrected, and native English revision is required.

  • This manuscript has received English editing by an accredited English editorial agency. We also have reconfirmed punctuation, and spelling et al.

POINT 6

Strength: in general, the study is easy to follow and well summarized, raising the main issues related to the topic. Limitation: although the main issues related to the topic raised by the authors, the report is not well interpreted in terms of the issue raised and the purpose/aim of the review is not clearly illustrated, related to other literature on the topic not well mentioned, in particular, recent studies have not taken into consideration.

  • The purpose of the study was presented. Please refer to the red color sentence on page 3, lines 82-85.
  • In this study, we would like to inform that the use of telemedicine is not for the purpose of blocking infectious diseases, such as COVID-19 et al, but rather for improving access to medical care for chronically ill patients in rural areas that are alienated from medical services.

Reviewer 2 Report

Thank you for allowing reviewing this interesting work. Authors presented a telemedicine model in the management of chronic disease patients. The study is well developed and before its acceptance, I would like to suggest some changes during the revision.

  1. The introduction is well structured. However, based on recent pandemic incidents, I recommend authors to mention the latest telemedicine developments for remote area patients in the introduction. You can consider the following suggestions     

Mittal M, Battineni G, Goyal LM, Chhetri B, Oberoi SV, Chintalapudi N, Amenta F. Cloud-based framework to mitigate the impact of COVID-19 on seafarers' mental health. Int Marit Health. 2020;71(3):213-214. doi: 10.5603/IMH.2020.0038.

Vidal-Alaball J, Acosta-Roja R, Pastor Hernández N, Sanchez Luque U, Morrison D, Narejos Pérez S, Perez-Llano J, Salvador Vèrges A, López Seguí F. Telemedicine in the face of the COVID-19 pandemic. Aten Primaria. 2020 Jun-Jul;52(6):418-422. doi: 10.1016/j.aprim.2020.04.003.

  1. The data that selected is secondary data, could you insert the repository source
  2. In section 2.2.2. The questions that are presented in the tool are they standard or tailor-made type? How can they validate, more details on this are missing? Please elaborate
  3. In the results section, authors have found any correlation between the subject characteristics like gender, employability, health security and prevalence of diseases?
  4. Some sentences are hard to follow, if possible, please conduct final English revision before submit

All the above, this manuscript has presented novel ideas of telemedicine that are in great need of present health communities.  

Author Response

Red letters in the manuscript indicate revised sentence.

Thank you for allowing reviewing this interesting work. Authors presented a telemedicine model in the management of chronic disease patients. The study is well developed and before its acceptance, I would like to suggest some changes during the revision.

  1. The introduction is well structured. However, based on recent pandemic incidents, I recommend authors to mention the latest telemedicine developments for remote area patients in the introduction. You can consider the following suggestions     

Mittal M, Battineni G, Goyal LM, Chhetri B, Oberoi SV, Chintalapudi N, Amenta F. Cloud-based framework to mitigate the impact of COVID-19 on seafarers' mental health. Int Marit Health. 2020;71(3):213-214. doi: 10.5603/IMH.2020.0038.

Vidal-Alaball J, Acosta-Roja R, Pastor Hernández N, Sanchez Luque U, Morrison D, Narejos Pérez S, Perez-Llano J, Salvador Vèrges A, López Seguí F. Telemedicine in the face of the COVID-19 pandemic. Aten Primaria. 2020 Jun-Jul;52(6):418-422. doi: 10.1016/j.aprim.2020.04.003.

  • We revised according to reviewer’s comment. Please refer to the red color sentence on page 10, lines 309 and page 12, reference 37.
  • In this study, we would like to inform that the use of telemedicine is not for the purpose of blocking infectious diseases, such as COVID-19 et al, but rather for improving access to medical care for chronically ill patients in rural areas that are alienated from medical services.

  1. The data that selected is secondary data, could you insert the repository source
  • The souces of the data is presented on page 3, line 96.

  1. In section 2.2.2. The questions that are presented in the tool are they standard or tailor-made type? How can they validate, more details on this are missing? Please elaborate
  • The items of MMS are standardized type. Please refer to the red color sentence on page 4, lines 143.
  • Details on the development process and validity evaluation of this tool are specified in the report of the Case Management Society of America.

  1. In the results section, authors have found any correlation between the subject characteristics like gender, employability, health security and prevalence of diseases?
  • We revised according to reviewer’s comment. Please refer to the red color sentence on page 5, lines 184-188.

  1. Some sentences are hard to follow, if possible, please conduct final English revision before submit
  • This manuscript has received English editing by an accredited English editorial agency. We also have reconfirmed punctuation, and spelling et al.
